# Supervised Learning for Dynamical System Learning

**Ahmed Hefny** *
Carnegie Mellon University
Pittsburgh, PA 15213
ahefny@cs.cmu.edu

**Carlton Downey** †
Carnegie Mellon University
Pittsburgh, PA 15213
cmdowney@cs.cmu.edu

**Geoffrey J. Gordon** ‡
Carnegie Mellon University
Pittsburgh, PA 15213
ggordon@cs.cmu.edu

## Abstract

Recently there has been substantial interest in spectral methods for learning dynamical systems. These methods are popular since they often offer a good tradeoff between computational and statistical efficiency. Unfortunately, they can be difficult to use and extend in practice: e.g., they can make it difficult to incorporate prior information such as sparsity or structure. To address this problem, we present a new view of dynamical system learning: we show how to learn dynamical systems by solving a sequence of ordinary supervised learning problems, thereby allowing users to incorporate prior knowledge via standard techniques such as $L_1$ regularization. Many existing spectral methods are special cases of this new framework, using linear regression as the supervised learner. We demonstrate the effectiveness of our framework by showing examples where nonlinear regression or lasso let us learn better state representations than plain linear regression does; the correctness of these instances follows directly from our general analysis.

## 1    Introduction

Likelihood-based approaches to learning dynamical systems, such as EM [1] and MCMC [2], can be slow and suffer from local optima. This difficulty has resulted in the development of so-called "spectral algorithms" [3], which rely on factorization of a matrix of observable moments; these algorithms are often fast, simple, and globally optimal.

Despite these advantages, spectral algorithms fall short in one important aspect compared to EM and MCMC: the latter two methods are meta-algorithms or frameworks that offer a clear template for developing new instances incorporating various forms of prior knowledge. For spectral algorithms, by contrast, there is no clear template to go from a set of probabilistic assumptions to an algorithm. In fact, researchers often relax model assumptions to make the algorithm design process easier, potentially discarding valuable information in the process.

To address this problem, we propose a new framework for dynamical system learning, using the idea of instrumental-variable regression [4, 5] to transform dynamical system learning to a sequence of ordinary supervised learning problems. This transformation allows us to apply the rich literature on supervised learning to incorporate many types of prior knowledge. Our new methods subsume a variety of existing spectral algorithms as special cases.

The remainder of this paper is organized as follows: first we formulate the new learning framework (Sec. 2). We then provide theoretical guarantees for the proposed methods (Sec. 4). Finally, we give

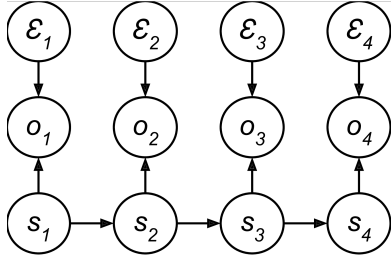

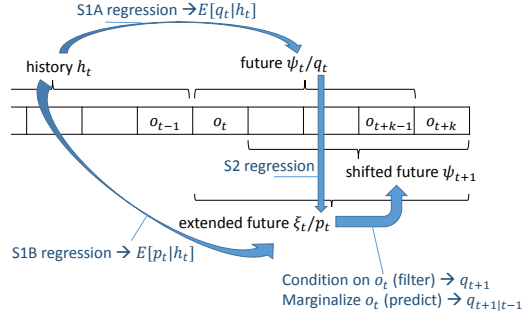

Figure 1: A latent-state dynamical system. Observation $o_t$ is determined by latent state $s_t$ and noise $\epsilon_t$.

Figure 2: Learning and applying a dynamical system with instrumental regression. The predictions from S1 provide training data to S2. At test time, we filter or predict using the weights from S2.

two examples of how our techniques let us rapidly design new and useful dynamical system learning methods by encoding modeling assumptions (Sec. 5).

## 2   A framework for spectral algorithms

A dynamical system is a stochastic process (i.e., a distribution over sequences of observations) such that, at any time, the distribution of future observations is fully determined by a vector $s_t$ called the *latent state*. The process is specified by three distributions: the initial state distribution $P(s_1)$, the state transition distribution $P(s_{t+1} \mid s_t)$, and the observation distribution $P(o_t \mid s_t)$. For later use, we write the observation $o_t$ as a function of the state $s_t$ and random noise $\epsilon_t$, as shown in Figure 1.

Given a dynamical system, one of the fundamental tasks is to perform inference, where we predict future observations given a history of observations. Typically this is accomplished by maintaining a distribution or *belief* over states $b_{t|t-1} = P(s_t \mid o_{1:t-1})$ where $o_{1:t-1}$ denotes the first $t-1$ observations. $b_{t|t-1}$ represents both our knowledge and our uncertainty about the true state of the system. Two core inference tasks are *filtering* and *prediction*.[1] In filtering, given the current belief $b_t = b_{t|t-1}$ and a new observation $o_t$, we calculate an updated belief $b_{t+1} = b_{t+1|t}$ that incorporates $o_t$. In prediction, we project our belief into the future: given a belief $b_{t|t-1}$ we estimate $b_{t+k|t-1} = P(s_{t+k} \mid o_{1:t-1})$ for some $k > 0$ (without incorporating any intervening observations).

The typical approach for learning a dynamical system is to explicitly learn the initial, transition, and observation distributions by maximum likelihood. Spectral algorithms offer an alternate approach to learning: they instead use the method of moments to set up a system of equations that can be solved in closed form to recover estimates of the desired parameters. In this process, they typically factorize a matrix or tensor of observed moments—hence the name "spectral."

Spectral algorithms often (but not always [6]) avoid explicitly estimating the latent state or the initial, transition, or observation distributions; instead they recover *observable operators* that can be used to perform filtering and prediction directly. To do so, they use an observable representation: instead of maintaining a belief $b_t$ over states $s_t$, they maintain the expected value of a sufficient statistic of future observations. Such a representation is often called a *(transformed) predictive state* [7].

In more detail, we define $q_t = q_{t|t-1} = \mathbb{E}[\psi_t \mid o_{1:t-1}]$, where $\psi_t = \psi(o_{t:t+k-1})$ is a vector of *future features*. The features are chosen such that $q_t$ determines the distribution of future observations

$P(o_{t:t+k-1} \mid o_{1:t-1})$.[2] Filtering then becomes the process of mapping a predictive state $q_t$ to $q_{t+1}$ conditioned on $o_t$, while prediction maps a predictive state $q_t = q_{t|t-1}$ to $q_{t+k|t-1} = \mathbb{E}[\psi_{t+k} \mid o_{1:t-1}]$ without intervening observations.

A typical way to derive a spectral method is to select a set of moments involving $\psi_t$, work out the expected values of these moments in terms of the observable operators, then invert this relationship to get an equation for the observable operators in terms of the moments. We can then plug in an empirical estimate of the moments to compute estimates of the observable operators.

While effective, this approach can be statistically inefficient (the goal of being able to solve for the observable operators is in conflict with the goal of maximizing statistical efficiency) and can make it difficult to incorporate prior information (each new source of information leads to new moments and a different and possibly harder set of equations to solve). To address these problems, we show that we can instead learn the observable operators by solving three supervised learning problems.

The main idea is that, just as we can represent a belief about a latent state $s_t$ as the conditional expectation of a vector of observable statistics, we can also represent any other distributions needed for prediction and filtering via their own vectors of observable statistics. Given such a representation, we can learn to filter and predict by learning how to map these vectors to one another.

In particular, the key intermediate quantity for filtering is the "extended and marginalized" belief $P(o_t, s_{t+1} \mid o_{1:t-1})$—or equivalently $P(o_{t:t+k} \mid o_{1:t-1})$. We represent this distribution via a vector $\xi_t = \xi(o_{t:t+k})$ of *features of the extended future*. The features are chosen such that the *extended state* $p_t = \mathbb{E}[\xi_t \mid o_{1:t-1}]$ determines $P(o_{t:t+k} \mid o_{1:t-1})$. Given $P(o_{t:t+k} \mid o_{1:t-1})$, filtering and prediction reduce respectively to conditioning on and marginalizing over $o_t$.

In many models (including Hidden Markov Models (HMMs) and Kalman filters), the extended state $p_t$ is linearly related to the predictive state $q_t$—a property we exploit for our framework. That is,

$$p_t = W q_t \tag{1}$$

for some linear operator $W$. For example, in a discrete system $\psi_t$ can be an indicator vector representing the joint assignment of the next $k$ observations, and $\xi_t$ can be an indicator vector for the next $k+1$ observations. The matrix $W$ is then the conditional probability table $P(o_{t:t+k} \mid o_{t:t+k-1})$.

Our goal, therefore, is to learn this mapping $W$. Naïvely, we might try to use linear regression for this purpose, substituting samples of $\psi_t$ and $\xi_t$ in place of $q_t$ and $p_t$ since we cannot observe $q_t$ or $p_t$ directly. Unfortunately, due to the overlap between observation windows, the noise terms on $\psi_t$ and $\xi_t$ are correlated. So, naïve linear regression will give a biased estimate of $W$.

To counteract this bias, we employ instrumental regression [4, 5]. Instrumental regression uses *instrumental variables* that are correlated with the input $q_t$ but not with the noise $\epsilon_{t:t+k}$. This property provides a criterion to denoise the inputs and outputs of the original regression problem: we remove that part of the input/output that is not correlated with the instrumental variables. In our case, since past observations $o_{1:t-1}$ do not overlap with future or extended future windows, they are not correlated with the noise $\epsilon_{t:t+k+1}$, as can be seen in Figure 1. Therefore, we can use *history features* $h_t = h(o_{1:t-1})$ as instrumental variables.

In more detail, by taking the expectation of (1) given $h_t$, we obtain an instrument-based moment condition: for all $t$,

$$\mathbb{E}[p_t \mid h_t] = \mathbb{E}[W q_t \mid h_t]$$
$$\mathbb{E}[\mathbb{E}[\xi_t \mid o_{1:t-1}] \mid h_t] = W \mathbb{E}[\mathbb{E}[\psi_t \mid o_{1:t-1}] \mid h_t]$$
$$\mathbb{E}[\xi_t \mid h_t] = W \mathbb{E}[\psi_t \mid h_t] \tag{2}$$

Assuming that there are enough independent dimensions in $h_t$ that are correlated with $q_t$, we maintain the rank of the moment condition when moving from (1) to (2), and we can recover $W$ by least squares regression if we can compute $\mathbb{E}[\psi_t \mid h_t]$ and $\mathbb{E}[\xi_t \mid h_t]$ for sufficiently many examples $t$.

Fortunately, conditional expectations such as $\mathbb{E}[\psi_t \mid h_t]$ are exactly what supervised learning algorithms are designed to compute. So, we arrive at our learning framework: we first use supervised

| Model/Algorithm | future features $\psi_t$ | extended future features $\xi_t$ | $f_{\text{filter}}$ |
|---|---|---|---|
| Spectral Algorithm for HMM [3] | $U^\top e_{o_t}$ where $e_{o_t}$ is an indicator vector and $U$ spans the range of $q_t$ (typically the top $m$ left singular vectors of the joint probability table $P(o_{t+1}, o_t)$) | $U^\top e_{o_{t+1}} \otimes e_{o_t}$ | Estimate a state normalizer from S1A output states. |
| SSID for Kalman filters (time dependent gain) | $x_t$ and $x_t \otimes x_t$, where $x_t = U^\top o_{t:t+k-1}$ for a matrix $U$ that spans the range of $q_t$ (typically the top $m$ left singular vectors of the covariance matrix $\text{Cov}(o_{t:t+k-1}, o_{t-k:t-1})$) | $y_t$ and $y_t \otimes y_t$, where $y_t$ is formed by stacking $U^\top o_{t+1:t+k}$ and $o_t$. | $p_t$ specifies a Gaussian distribution where conditioning on $o_t$ is straightforward. |
| SSID for stable Kalman filters (constant gain) | $U^\top o_{t:t+k-1}$ ($U$ obtained as above) | $o_t$ and $U^\top o_{t+1:t+k}$ | Estimate steady-state covariance by solving Riccati equation [8]. $p_t$ together with the steady-state covariance specify a Gaussian distribution where conditioning on $o_t$ is straightforward. |
| Uncontrolled HSE-PSR [9] | Evaluation functional $k_s(o_{t:t+k-1}, .)$ for a characteristic kernel $k_s$ | $k_o(o_t, .) \otimes k_o(o_t, .)$ and $\psi_{t+1} \otimes k_o(o_t, .)$ | Kernel Bayes rule [10]. |

Table 1: Examples of existing spectral algorithms reformulated as two-stage instrument regression with linear S1 regression. Here $o_{t_1:t_2}$ is a vector formed by stacking observations $o_{t_1}$ through $o_{t_2}$ and $\otimes$ denotes the outer product. Details and derivations can be found in the supplementary material.

learning to estimate $\mathbb{E}[\psi_t \mid h_t]$ and $\mathbb{E}[\xi_t \mid h_t]$, effectively *denoising* the training examples, and then use these estimates to compute $W$ by finding the least squares solution to (2).

In summary, learning and inference of a dynamical system through instrumental regression can be described as follows:

- **Model Specification:** Pick features of history $h_t = h(o_{1:t-1})$, future $\psi_t = \psi(o_{t:t+k-1})$ and extended future $\xi_t = \xi(o_{t:t+k})$. $\psi_t$ must be a sufficient statistic for $\mathbb{P}(o_{t:t+k-1} \mid o_{1:t-1})$. $\xi_t$ must satisfy

  - $\mathbb{E}[\psi_{t+1} \mid o_{1:t-1}] = f_{\text{predict}}(\mathbb{E}[\xi_t \mid o_{1:t-1}])$ for a known function $f_{\text{predict}}$.
  - $\mathbb{E}[\psi_{t+1} \mid o_{1:t}] = f_{\text{filter}}(\mathbb{E}[\xi_t \mid o_{1:t-1}], o_t)$ for a known function $f_{\text{filter}}$.

- **S1A (Stage 1A) Regression:** Learn a (possibly non-linear) regression model to estimate $\bar{\psi}_t = \mathbb{E}[\psi_t \mid h_t]$. The training data for this model are $(h_t, \psi_t)$ across time steps $t$.[3]
- **S1B Regression:** Learn a (possibly non-linear) regression model to estimate $\bar{\xi}_t = \mathbb{E}[\xi_t \mid h_t]$. The training data for this model are $(h_t, \xi_t)$ across time steps $t$.
- **S2 Regression:** Use the feature expectations estimated in S1A and S1B to train a model to predict $\bar{\xi}_t = W\bar{\psi}_t$, where $W$ is a linear operator. The training data for this model are estimates of $(\bar{\psi}_t, \bar{\xi}_t)$ obtained from S1A and S1B across time steps $t$.
- **Initial State Estimation:** Estimate an initial state $q_1 = \mathbb{E}[\psi_1]$ by averaging $\psi_1$ across several example realizations of our time series.[4]
- **Inference:** Starting from the initial state $q_1$, we can maintain the predictive state $q_t = \mathbb{E}[\psi_t \mid o_{1:t-1}]$ through filtering: given $q_t$ we compute $p_t = \mathbb{E}[\xi_t \mid o_{1:t-1}] = Wq_t$. Then, given the observation $o_t$, we can compute $q_{t+1} = f_{\text{filter}}(p_t, o_t)$. Or, in the absence of $o_t$, we can predict the next state $q_{t+1|t-1} = f_{\text{predict}}(p_t)$. Finally, by definition, the predictive state $q_t$ is sufficient to compute $\mathbb{P}(o_{t:t+k-1} \mid o_{1:t-1})$.[5]

The process of learning and inference is depicted in Figure 2. Modeling assumptions are reflected in the choice of the statistics $\psi$, $\xi$ and $h$ as well as the regression models in stages S1A and S1B. Table 1 demonstrates that we can recover existing spectral algorithms for dynamical system learning using linear S1 regression. In addition to providing a unifying view of some successful learning algorithms, the new framework also paves the way for extending these algorithms in a theoretically justified manner, as we demonstrate in the experiments below.

## 3 Related Work

This work extends predictive state learning algorithms for dynamical systems, which include spectral algorithms for Kalman filters [11], Hidden Markov Models [3, 12], Predictive State Representations (PSRs) [13, 14] and Weighted Automata [15]. It also extends kernel variants such as [9], which builds on [16]. All of the above work effectively uses linear regression or linear ridge regression (although not always in an obvious way).

One common aspect of predictive state learning algorithms is that they exploit the covariance structure between future and past observation sequences to obtain an unbiased observable state representation. Boots and Gordon [17] note the connection between this covariance and (linear) instrumental regression in the context of the HSE-HMM. We use this connection to build a general framework for dynamical system learning where the state space can be identified using arbitrary (possibly nonlinear) supervised learning methods. This generalization lets us incorporate prior knowledge to learn compact or regularized models; our experiments demonstrate that this flexibility lets us take better advantage of limited data.

Reducing the problem of learning dynamical systems with latent state to supervised learning bears similarity to Langford et al.'s sufficient posterior representation (SPR) [18], which encodes the state by the sufficient statistics of the conditional distribution of the next observation and represents system dynamics by three vector-valued functions that are estimated using supervised learning approaches. While SPR allows all of these functions to be non-linear, it involves a rather complicated training procedure involving multiple iterations of model refinement and model averaging, whereas our framework only requires solving three regression problems in sequence. In addition, the theoretical analysis of [18] only establishes the consistency of SPR learning assuming that all regression steps are solved perfectly. Our work, on the other hand, establishes convergence rates based on the performance of S1 regression.

## 4 Theoretical Analysis

In this section we present error bounds for two-stage instrumental regression. These bounds hold regardless of the particular S1 regression method used, assuming that the S1 predictions converge to the true conditional expectations. The bounds imply that our overall method is consistent.

Let $(x_t, y_t, z_t) \in (\mathcal{X}, \mathcal{Y}, \mathcal{Z})$ be i.i.d. triplets of input, output, and instrumental variables. (Lack of independence will result in slower convergence in proportion to the mixing time of our process.) Let $\bar{x}_t$ and $\bar{y}_t$ denote $\mathbb{E}[x_t \mid z_t]$ and $\mathbb{E}[y_t \mid z_t]$. And, let $\hat{x}_t$ and $\hat{y}_t$ denote $\hat{\mathbb{E}}[x_t \mid z_t]$ and $\hat{\mathbb{E}}[y_t \mid z_t]$ as estimated by the S1A and S1B regression steps. Here $\bar{x}_t, \hat{x}_t \in \mathcal{X}$ and $\bar{y}_t, \hat{y}_t \in \mathcal{Y}$.

We want to analyze the convergence of the output of S2 regression—that is, of the weights $W$ given by ridge regression between S1A outputs and S1B outputs:

$$\hat{W}_\lambda = \left( \sum_{t=1}^{T} \hat{y}_t \otimes \hat{x}_t \right) \left( \sum_{t=1}^{T} \hat{x}_t \otimes \hat{x}_t + \lambda I_{\mathcal{X}} \right)^{-1} \tag{3}$$

Here $\otimes$ denotes tensor (outer) product, and $\lambda > 0$ is a regularization parameter that ensures the invertibility of the estimated covariance.

Before we state our main theorem we need to quantify the quality of S1 regression in a way that is independent of the S1 functional form. To do so, we place a bound on the S1 error, and assume that this bound converges to zero: given the definition below, for each fixed $\delta$, $\lim_{N \to \infty} \eta_{\delta,N} = 0$.

**Definition 1** (S1 Regression Bound). *For any $\delta > 0$ and $N \in \mathbb{N}^+$, the S1 regression bound $\eta_{\delta,N} > 0$ is a number such that, with probability at least $(1 - \delta/2)$, for all $1 \leq t \leq N$:*

$$\|\hat{x}_t - \bar{x}_t\|_{\mathcal{X}} < \eta_{\delta,N}$$
$$\|\hat{y}_t - \bar{y}_t\|_{\mathcal{Y}} < \eta_{\delta,N}$$

In many applications, $\mathcal{X}$, $\mathcal{Y}$ and $\mathcal{Z}$ will be finite dimensional real vector spaces: $\mathbb{R}^{d_x}$, $\mathbb{R}^{d_y}$ and $\mathbb{R}^{d_z}$. However, for generality we state our results in terms of arbitrary reproducing kernel Hilbert spaces. In this case S2 uses kernel ridge regression, leading to methods such as HSE-PSRs. For

this purpose, let $\Sigma_{\bar{x}\bar{x}}$ and $\Sigma_{\bar{y}\bar{y}}$ denote the (uncentered) covariance operators of $\bar{x}$ and $\bar{y}$ respectively: $\Sigma_{\bar{x}\bar{x}} = \mathbb{E}[\bar{x} \otimes \bar{x}]$, $\Sigma_{\bar{y}\bar{y}} = \mathbb{E}[\bar{y} \otimes \bar{y}]$. And, let $\overline{\mathcal{R}(\Sigma_{\bar{x}\bar{x}})}$ denote the closure of the range of $\Sigma_{\bar{x}\bar{x}}$.

With the above assumptions, Theorem 2 gives a generic error bound on S2 regression in terms of S1 regression. If $\mathcal{X}$ and $\mathcal{Y}$ are finite dimensional and $\Sigma_{\bar{x}\bar{x}}$ has full rank, then using ordinary least squares (i.e., setting $\lambda = 0$) will give the same bound, but with $\lambda$ in the first two terms replaced by the minimum eigenvalue of $\Sigma_{\bar{x}\bar{x}}$, and the last term dropped.

**Theorem 2.** *Assume that* $\|\bar{x}\|_{\mathcal{X}}, \|\bar{x}\|_{\mathcal{Y}} < c < \infty$ *almost surely. Assume $W$ is a Hilbert-Schmidt operator, and let $\hat{W}_\lambda$ be as defined in* (3). *Then, with probability at least* $1 - \delta$, *for each* $x_{\text{test}} \in \overline{\mathcal{R}(\Sigma_{\bar{x}\bar{x}})}$ *s.t.* $\|x_{\text{test}}\|_{\mathcal{X}} \leq 1$, *the error* $\|\hat{W}_\lambda x_{\text{test}} - W x_{\text{test}}\|_{\mathcal{Y}}$ *is bounded by*

$$\underbrace{O\left(\eta_{\delta,N}\left(\frac{1}{\lambda} + \frac{\sqrt{1 + \sqrt{\frac{\log(1/\delta)}{N}}}}{\lambda^{\frac{3}{2}}}\right)\right)}_{\text{error in S1 regression}} + \underbrace{O\left(\frac{\log(1/\delta)}{\sqrt{N}}\left(\frac{1}{\lambda} + \frac{1}{\lambda^{\frac{3}{2}}}\right)\right)}_{\text{error from finite samples}} + \underbrace{O\left(\sqrt{\lambda}\right)}_{\text{error from regularization}}$$

We defer the proof to the supplementary material. The supplementary material also provides explicit finite-sample bounds (including expressions for the constants hidden by $O$-notation), as well as concrete examples of S1 regression bounds $\eta_{\delta,N}$ for practical regression models.

Theorem 2 assumes that $x_{\text{test}}$ is in $\overline{\mathcal{R}(\Sigma_{\bar{x}\bar{x}})}$. For dynamical systems, all valid states satisfy this property. However, with finite data, estimation errors may cause the estimated state $\hat{q}_t$ (i.e., $x_{\text{test}}$) to have a non-zero component in $\mathcal{R}^\perp(\Sigma_{\bar{x}\bar{x}})$. Lemma 3 bounds the effect of such errors: it states that, in a stable system, this component gets smaller as S1 regression performs better. The main limitation of Lemma 3 is the assumption that $f_{\text{filter}}$ is $L$-Lipchitz, which essentially means that the model's estimated probability for $o_t$ is bounded below. There is no way to guarantee this property in practice; so, Lemma 3 provides suggestive evidence rather than a guarantee that our learned dynamical system will predict well.

**Lemma 3.** *For observations $o_{1:T}$, let $\hat{q}_t$ be the estimated state given $o_{1:t-1}$. Let $\tilde{q}_t$ be the projection of $\hat{q}_t$ onto $\overline{\mathcal{R}(\Sigma_{\bar{x}\bar{x}})}$. Assume $f_{\text{filter}}$ is $L$-Lipchitz on $p_t$ when evaluated at $o_t$, and $f_{\text{filter}}(p_t, o_t) \in \overline{\mathcal{R}(\Sigma_{\bar{x}\bar{x}})}$ for any $p_t \in \overline{\mathcal{R}(\Sigma_{\bar{y}\bar{y}})}$. Given the assumptions of theorem 2 and assuming that $\|\hat{q}_t\|_{\mathcal{X}} \leq R$ for all $1 \leq t \leq T$, the following holds for all $1 \leq t \leq T$ with probability at least $1 - \delta/2$.*

$$\|\epsilon_t\|_{\mathcal{X}} = \|\hat{q}_t - \tilde{q}_t\|_{\mathcal{X}} = O\left(\frac{\eta_{\delta,N}}{\sqrt{\lambda}}\right)$$

Since $\hat{W}_\lambda$ is bounded, the prediction error due to $\epsilon_t$ diminishes at the same rate as $\|\epsilon_t\|_{\mathcal{X}}$.

## 5 Experiments and Results

We now demonstrate examples of tweaking the S1 regression to gain advantage. In the first experiment we show that nonlinear regression can be used to reduce the number of parameters needed in S1, thereby improving statistical performance for learning an HMM. In the second experiment we show that we can encode prior knowledge as regularization.

### 5.1 Learning A Knowledge Tracing Model

In this experiment we attempt to model and predict the performance of students learning from an interactive computer-based tutor. We use the Bayesian knowledge tracing (BKT) model [19], which is essentially a 2-state HMM: the state $s_t$ represents whether a student has learned a knowledge component (KC), and the observation $o_t$ represents the success/failure of solving the $t^{\text{th}}$ question in a sequence of questions that cover this KC. Figure 3 summarizes the model. The events denoted by guessing, slipping, learning and forgetting typically have relatively low probabilities.

### 5.1.1 Data Description

We evaluate the model using the "Geometry Area (1996-97)" data available from DataShop [20]. This data was generated by students learning introductory geometry, and contains attempts by 59

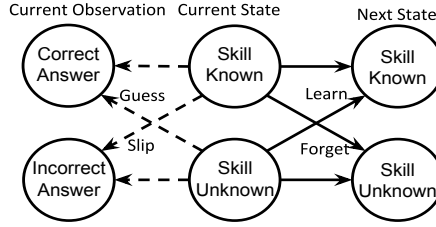

Figure 3: Transitions and observations in BKT. Each node represents a possible *value* of the state or observation. Solid arrows represent transitions while dashed arrows represent observations.

students in 12 knowledge components. As is typical for BKT, we consider a student's attempt at a question to be correct iff the student entered the correct answer on the first try, without requesting any hints from the help system. Each training sequence consists of a sequence of first attempts for a student/KC pair. We discard sequences of length less than 5, resulting in a total of 325 sequences.

### 5.1.2 Models and Evaluation

Under the (reasonable) assumption that the two states have distinct observation probabilities, this model is 1-observable. Hence we define the predictive state to be the expected next observation, which results in the following statistics: $\psi_t = o_t$ and $\xi_t = o_t \otimes_k o_{t+1}$, where $o_t$ is represented by a 2 dimensional indicator vector and $\otimes_k$ denotes the Kronecker product. Given these statistics, the extended state $p_t = \mathbb{E}[\xi_t \mid o_{1:t-1}]$ is a joint probability table of $o_{t:t+1}$.

We compare three models that differ by history features and S1 regression method:

**Spec-HMM:** This baseline uses $h_t = o_{t-1}$ and linear S1 regression, making it equivalent to the spectral HMM method of [3], as detailed in the supplementary material.

**Feat-HMM:** This baseline represents $h_t$ by an indicator vector of the joint assignment of the previous $b$ observations (we set $b$ to 4) and uses linear S1 regression. This is essentially a feature-based spectral HMM [12]. It thus incorporates more history information compared to Spec-HMM at the expense of increasing the number of S1 parameters by $O(2^b)$.

**LR-HMM:** This model represents $h_t$ by a binary vector of length $b$ encoding the previous $b$ observations and uses logistic regression as the S1 model. Thus, it uses the same history information as Feat-HMM but reduces the number of parameters to $O(b)$ at the expense of inductive bias.

We evaluated the above models using 1000 random splits of the 325 sequences into 200 training and 125 testing. For each testing observation $o_t$ we compute the absolute error between actual and expected value (i.e. $|\delta_{o_t=1} - \hat{P}(o_t = 1 \mid o_{1:t-1})|$). We report the mean absolute error for each split. The results are displayed in Figure 4.[6] We see that, while incorporating more history information increases accuracy (Feat-HMM vs. Spec-HMM), being able to incorporate the same information using a more compact model gives an additional gain in accuracy (LR-HMM vs. Feat-HMM). We also compared the LR-HMM method to an HMM trained using expectation maximization (EM). We found that the LR-HMM model is much faster to train than EM while being on par with it in terms of prediction error.[7]

### 5.2 Modeling Independent Subsystems Using Lasso Regression

Spectral algorithms for Kalman filters typically use the left singular vectors of the covariance between history and future features as a basis for the state space. However, this basis hides any sparsity that might be present in our original basis. In this experiment, we show that we can instead use lasso (without dimensionality reduction) as our S1 regression algorithm to discover sparsity. This is useful, for example, when the system consists of multiple independent subsystems, each of which affects a subset of the observation coordinates.

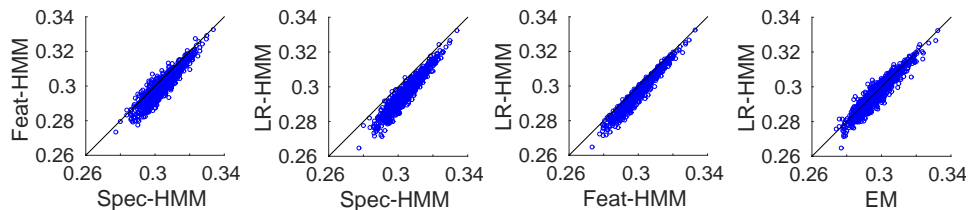

| Model | Spec-HMM | Feat-HMM | LR-HMM | EM |
|---|---|---|---|---|
| Training time (relative to Spec-HMM) | 1 | 1.02 | 2.219 | 14.323 |

Figure 4: Experimental results: each graph compares the performance of two models (measured by mean absolute error) on 1000 train/test splits. The black line is $x = y$. Points below this line indicate that model $y$ is better than model $x$. The table shows training time.

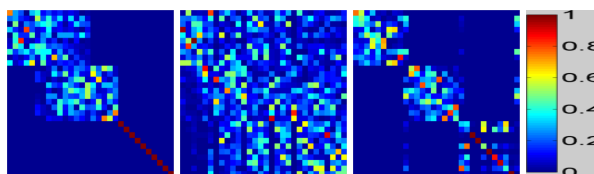

Figure 5: Left singular vectors of (left) true linear predictor from $o_{t-1}$ to $o_t$ (i.e. $OTO^+$), (middle) covariance matrix between $o_t$ and $o_{t-1}$ and (right) S1 Sparse regression weights. Each column corresponds to a singular vector (only absolute values are depicted). Singular vectors are ordered by their mean coordinate, interpreting absolute values as a probability distribution over coordinates.

To test this idea we generate a sequence of 30-dimensional observations from a Kalman filter. Observation dimensions 1 through 10 and 11 through 20 are generated from two independent subsystems of state dimension 5. Dimensions 21-30 are generated from white noise. Each subsystem's transition and observation matrices have random Gaussian coordinates, with the transition matrix scaled to have a maximum eigenvalue of 0.95. States and observations are perturbed by Gaussian noise with covariance of $0.01I$ and $1.0I$ respectively.

We estimate the state space basis using 1000 examples (assuming 1-observability) and compare the singular vectors of the past to future regression matrix to those obtained from the Lasso regression matrix. The result is shown in figure 5. Clearly, using Lasso as stage 1 regression results in a basis that better matches the structure of the underlying system.

## 6   Conclusion

In this work we developed a general framework for dynamical system learning using supervised learning methods. The framework relies on two key principles: first, we extend the idea of predictive state to include extended state as well, allowing us to represent all of inference in terms of predictions of observable features. Second, we use past features as instruments in an instrumental regression, denoising state estimates that then serve as training examples to estimate system dynamics.

We have shown that this framework encompasses and provides a unified view of some previous successful dynamical system learning algorithms. We have also demostrated that it can be used to extend existing algorithms to incorporate nonlinearity and regularizers, resulting in better state estimates. As future work, we would like to apply this framework to leverage additional techniques such as manifold embedding and transfer learning in stage 1 regression. We would also like to extend the framework to controlled processes.

## Footnotes

*This material is based upon work funded and supported by the Department of Defense under Contract No. FA8721-05-C-0003 with Carnegie Mellon University for the operation of the Software Engineering Institute, a federally funded research and development center.

†Supported by a grant from the PNC Center for Financial Services Innovation

‡Supported by NIH grant R01 MH 064537 and ONR contract N000141512365.

[1]There are other forms of inference in addition to filtering and prediction, such as smoothing and likelihood evaluation, but they are outside the scope of this paper.

[2]For convenience we assume that the system is $k$-observable: that is, the distribution of all future observations is determined by the distribution of the next $k$ observations. (Note: not by the next $k$ observations themselves.) At the cost of additional notation, this restriction could easily be lifted.

[3]Our bounds assume that the training time steps $t$ are sufficiently spaced for the underlying process to mix, but in practice, the error will only get smaller if we consider all time steps $t$.

[4]Assuming ergodicity, we can set the initial state to be the empirical average vector of future features in a single long sequence, $\frac{1}{T}\sum_{t=1}^{T}\psi_t$.

[5]It might seem reasonable to learn $q_{t+1} = f_{\text{combined}}(q_t, o_t)$ directly, thereby avoiding the need to separately estimate $p_t$ and condition on $o_t$. Unfortunately, $f_{\text{combined}}$ is nonlinear for common models such as HMMs.

[6]The differences have similar sign but smaller magnitude if we use RMSE instead of MAE.

[7]We used MATLAB's built-in logistic regression and EM functions.

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
