[Supplementary Material]

# Supervised Learning for Dynamical System Learning (Supplementary)

## A  Spectral and HSE Dynamical System Learning as Regression

In this section we provide examples of mapping some of the successful dynamical system learning algorithms to our framework.

### A.1  HMM

In this section we show that we can use instrumental regression framework to reproduce the spectral learning algorithm for learning HMM [1]. We consider 1-observable models but the argument applies to $k$-observable models. In this case we use $\psi_t = e_{o_t}$ and $\xi_t = e_{o_{t:t+1}} = e_{o_t} \otimes_k e_{o_{t+1}}$, where $\otimes_k$ denotes the kronecker product. Let $P_{i,j} \equiv \mathbb{E}[e_{o_i} \otimes e_{o_j}]$ be the joint probability table of observations $i$ and $j$ and let $\hat{P}_{i,j}$ be its estimate from the data. We start with the (very restrictive) case where $P_{1,2}$ is invertible. Given samples of $h_2 = e_{o_1}$, $\psi_2 = e_{o_2}$ and $\xi_2 = e_{o_{2:3}}$, in S1 regression we apply linear regression to learn two matrices $\hat{W}_{2,1}$ and $\hat{W}_{2:3,1}$ such that:

$$\hat{\mathbb{E}}[\psi_2|h_2] = \hat{\Sigma}_{o_2 o_1}\hat{\Sigma}_{o_1}^{-1}h_2 = \hat{P}_{2,1}\hat{P}_{1,1}^{-1}h_t \equiv \hat{W}_{2,1}h_2 \tag{A.1}$$

$$\hat{\mathbb{E}}[\xi_2|h_2] = \hat{\Sigma}_{o_{2:3}o_1}\hat{\Sigma}_{o_1}^{-1}h_2 = \hat{P}_{2:3,1}\hat{P}_{1,1}^{-1}h_2 \equiv \hat{W}_{2:3,1}h_2, \tag{A.2}$$

where $P_{2:3,1} \equiv \mathbb{E}[e_{o_{2:3}} \otimes e_{o_1}]$

In S2 regression, we learn the matrix $\hat{W}$ that gives the least squares solution to the system of equations

$$\hat{\mathbb{E}}[\xi_2|h_2] \equiv \hat{W}_{2:3,1}e_{o_1} = \hat{W}(\hat{W}_{2,1}e_{o_1}) \equiv \hat{W}\hat{\mathbb{E}}[\psi_2|h_2] \quad \text{, for given samples of } h_2$$

which gives

$$\begin{aligned}
\hat{W} &= \hat{W}_{2:3,1}\hat{\mathbb{E}}[e_{o_1}e_{o_1}^\top]\hat{W}_{2,1}^\top \left( \hat{W}_{2,1}\hat{\mathbb{E}}[e_{o_1}e_{o_1}^\top]\hat{W}_{2,1}^\top \right)^{-1} \\
&= \left( \hat{P}_{2:3,1}\hat{P}_{1,1}^{-1}\hat{P}_{2,1}^\top \right) \left( \hat{P}_{2,1}\hat{P}_{1,1}^{-1}\hat{P}_{2,1}^\top \right)^{-1} \\
&= \hat{P}_{2:3,1} \left( \hat{P}_{2,1} \right)^{-1}
\end{aligned} \tag{A.3}$$

Having learned the matrix $\hat{W}$, we can estimate

$$\hat{P}_t \equiv \hat{W}q_t$$

starting from a state $q_t$. Since $p_t$ specifies a joint distribution over $e_{o_{t+1}}$ and $e_{o_t}$ we can easily condition on (or marginalize $o_t$) to obtain $q_{t+1}$. We will show that this is equivalent to learning and applying observable operators as in [1]:

For a given value $x$ of $o_2$, define

$$B_x = u_x^\top \hat{W} = u_x^\top \hat{P}_{2:3,1} \left( \hat{P}_{2,1}^\top \right)^{-1}, \tag{A.4}$$

where $u_x$ is an $|\mathcal{O}| \times |\mathcal{O}|^2$ matrix which selects a block of rows in $\hat{P}_{2:3,1}$ corresponding to $o_2 = x$. Specifically, $u_x = \delta_x \otimes_k I_{|\mathcal{O}|}$. [1].

$$
\begin{aligned}
q_{t+1} = \hat{\mathbb{E}}[e_{o_{t+1}}|o_{1:t}] &\propto u_{o_t}^\top \hat{\mathbb{E}}[e_{o_{t:t+1}}|o_{1:t-1}] \\
&= u_{o_t}^\top \hat{\mathbb{E}}[\xi_t|o_{1:t-1}] = u_{o_t}^\top \hat{W}\mathbb{E}[\psi_t|o_{1:t-1}] = B_{o_t} q_t
\end{aligned}
$$

with a normalization constant given by

$$
\frac{1}{1^\top B_{o_t} q_t} \tag{A.5}
$$

Now we move to a more realistic setting, where we have $\operatorname{rank}(P_{2,1}) = m < |\mathcal{O}|$. Therefore we project the predictive state using a matrix $U$ that preserves the dynamics, by requiring that $U^\top O$ (i.e. $U$ is an independent set of columns spanning the range of the HMM observation matrix $O$).

It can be shown [1] that $\mathcal{R}(O) = \mathcal{R}(P_{2,1}) = \mathcal{R}(P_{2,1}P_{1,1}^{-1})$. Therefore, we can use the leading $m$ left singular vectors of $\hat{W}_{2,1}$, which corresponds to replacing the linear regression in S1A with a reduced rank regression. However, for the sake of our discussion we will use the singular vectors of $P_{2,1}$. In more detail, let $[U, S, V]$ be the rank-$m$ SVD decomposition of $P_{2,1}$. We use $\psi_t = U^\top e_{o_t}$ and $\xi_t = e_{o_t} \otimes_k U^\top e_{o_{t+1}}$. S1 weights are then given by $\hat{W}_{2,1}^{rr} = U^\top \hat{W}_{2,1}$ and $\hat{W}_{2:3,1}^{rr} = (I_{|\mathcal{O}|} \otimes_k U^\top)\hat{W}_{2:3,1}$ and S2 weights are given by

$$
\begin{aligned}
\hat{W}^{rr} &= (I_{|\mathcal{O}|} \otimes_k U^\top)\hat{W}_{2:3,1}\hat{\mathbb{E}}[e_{o_1}e_{o_1}^\top]\hat{W}_{2,1}^\top U \left(U^\top \hat{W}_{2,1}\hat{\mathbb{E}}[e_{o_1}e_{o_1}^\top]\hat{W}_{2,1}^\top U\right)^{-1} \\
&= (I_{|\mathcal{O}|} \otimes_k U^\top)\hat{P}_{2:3,1}\hat{P}_{1,1}^{-1}VS \left(SV^\top \hat{P}_{1,1}^{-1}VS\right)^{-1} \\
&= (I_{|\mathcal{O}|} \otimes_k U^\top)\hat{P}_{2:3,1}\hat{P}_{1,1}^{-1}V \left(V^\top \hat{P}_{1,1}^{-1}V\right)^{-1} S^{-1} \tag{A.6}
\end{aligned}
$$

In the limit of infinite data, $V$ spans $\operatorname{range}(O) = \operatorname{rowspace}(P_{2:3,1})$ and hence $P_{2:3,1} = P_{2:3,1}VV^\top$. Substituting in (A.6) gives

$$
W^{rr} = (I_{|\mathcal{O}|} \otimes_k U^\top)P_{2:3,1}VS^{-1} = (I_{|\mathcal{O}|} \otimes_k U^\top)P_{2:3,1} \left(U^\top P_{2,1}\right)^+
$$

Similar to the full-rank case we define, for each observation $x$ an $m \times |\mathcal{O}|^2$ selector matrix $u_x = \delta_x \otimes_k I_m$ and an observation operator

$$
B_x = u_x^\top \hat{W}^{rr} \rightarrow U^\top P_{3,x,1} \left(U^\top P_{2,1}\right)^+ \tag{A.7}
$$

This is exactly the observation operator obtained in [1]. However, instead of using A.6, they use A.7 with $P_{3,x,1}$ and $P_{2,1}$ replaced by their empirical estimates.

Note that for a state $b_t = \mathbb{E}[\psi_t|o_{1:t-1}]$, $B_x b_t = P(o_t|o_{1:t-1})\mathbb{E}[\psi_{t+1}|o_{1:t}] = P(o_t|o_{1:t-1})b_{t+1}$. To get $b_{t+1}$, the normalization constant becomes $\frac{1}{P(o_t|o_{1:t-1})} = \frac{1}{b_\infty^\top B_x b_t}$, where $b_\infty^\top b = 1$ for any valid predictive state $b$. To estimate $b_\infty$ we solve the aforementioned condition for states estimated from all possible values of history features $h_t$. This gives,

$$
b_\infty^\top \hat{W}_{2,1}^{rr} I_{|\mathcal{O}|} = b_\infty^\top U^\top \hat{P}_{2,1}\hat{P}_{1,1}^{-1}I_{|\mathcal{O}|} = 1_{|\mathcal{O}|}^\top,
$$

where the columns of $I_{|\mathcal{O}|}$ represent all possible values of $h_t$. This in turn gives

$$
\begin{aligned}
b_\infty^\top &= 1_{|\mathcal{O}|}^\top \hat{P}_{1,1}(U^\top \hat{P}_{2,1})^+ \\
&= \hat{P}_1^\top (U^\top \hat{P}_{2,1})^+,
\end{aligned}
$$

the same estimator proposed in [1].

## A.2 Stationary Kalman Filter

A Kalman filter is given by

$$s_t = Os_{t-1} + \nu_t$$
$$o_t = Ts_t + \epsilon_t$$
$$\nu_t \sim \mathcal{N}(0, \Sigma_s)$$
$$\epsilon_t \sim \mathcal{N}(0, \Sigma_o)$$

We consider the case of a *stationary* filter where $\Sigma_t \equiv \mathbb{E}[s_t s_t^\top]$ is independent of $t$. We choose our statistics

$$h_t = o_{t-H:t-1}$$
$$\psi_t = o_{t:t+F-1}$$
$$\xi_t = o_{t:t+F},$$

Where a window of observations is represented by stacking individual observations into a single vector. It can be shown [2, 3] that

$$\mathbb{E}[s_t|h_t] = \Sigma_{s,h}\Sigma_{h,h}^{-1}h_t$$

and it follows that

$$\mathbb{E}[\psi_t|h_t] = \Gamma\Sigma_{s,h}\Sigma_{h,h}^{-1}h_t = W_1 h_t$$
$$\mathbb{E}[\xi_t|h_t] = \Gamma_+\Sigma_{s,h}\Sigma_{h,h}^{-1}h_t = W_2 h_t$$

where $\Gamma$ is the extended observation operator

$$\Gamma \equiv \begin{pmatrix} O \\ OT \\ \vdots \\ OT^F \end{pmatrix}, \Gamma_+ \equiv \begin{pmatrix} O \\ OT \\ \vdots \\ OT^{F+1} \end{pmatrix}$$

It follows that $F$ and $H$ must be large enough to have $\mathrm{rank}(W) = n$. Let $U \in \mathbb{R}^{mF \times n}$ be the matrix of left singular values of $W_1$ corresponding to non-zero singular values. Then $U^\top\Gamma$ is invertible and we can write

$$\mathbb{E}[\psi_t|h_t] = UU^\top\Gamma\Sigma_{s,h}\Sigma_{h,h}^{-1}h_t = W_1 h_t$$
$$\mathbb{E}[\xi_t|h_t] = \Gamma_+\Sigma_{s,h}\Sigma_{h,h}^{-1}h_t = W_2 h_t$$
$$\mathbb{E}[\xi_t|h_t] = \Gamma_+(U^\top\Gamma)^{-1}U^\top\left(UU^\top\Gamma\Sigma_{s,h}\Sigma_{h,h}^{-1}h_t\right)$$
$$= W\mathbb{E}[\psi_t|h_t]$$

which matches the instrumental regression framework. For the steady-state case (constant Kalman gain), one can estimate $\Sigma_\xi$ given the data and the parameter $W$ by solving Riccati equation as described in [3]. $\mathbb{E}[\xi_t|o_{1:t-1}]$ and $\Sigma_\xi$ then specify a joint Gaussian distribution over the next $F+1$ observations where marginalization and conditioning can be easily performed.

We can also assume a Kalman filter that is not in the steady state (i.e. the Kalman gain is not constant). In this case we need to maintain sufficient statistics for a predictive Gaussian distribution (i.e. mean and covariance). Let vec denote the vectorization operation, which stacks the columns of a matrix into a single vector. We can stack $h_t$ and $\mathrm{vec}(h_t h_t^\top)$ to into a single vector that we refer to as 1st+2nd moments vector. We do the same for future and extended future. We can, in principle, perform linear regression on these 1st+2nd moment vectors but that requires an unnecessarily large number of parameters. Instead, we can learn an S1A regression function of the form

$$\mathbb{E}[\psi_t|h_t] = W_1 h_t \tag{A.8}$$
$$\mathbb{E}[\psi_t\psi_t^\top|h_t] = W_1 h_t h_t^\top W_1 + R \tag{A.9}$$
$$\tag{A.10}$$

Where $R$ is simply the covariance of the residuals of the 1st moment regression (i.e. covariance of $r_t = \psi_t - \mathbb{E}[\psi_t|h_t]$). This is still a linear model in terms of 1st+2nd moment vectors and hence we can do the same for S1B and S2 regression models. This way, the extended belief vector $p_t$ (the expectation of 1st+2nd moments of extended future) fully specifies a joint distribution over the next $F + 1$ observations.

## A.3 HSE-PSR

We define a class of non-parametric two-stage instrumental regression models. By using conditional mean embedding [4] as S1 regression model, we recover a single-action variant of HSE-PSR [5]. Let $\mathcal{X}, \mathcal{Y}, \mathcal{Z}$ denote three reproducing kernel Hilbert spaces with reproducing kernels $k_\mathcal{X}, k_\mathcal{Y}$ and $k_\mathcal{Z}$ respectively. Assume $\psi_t \in \mathcal{X}$ and that $\xi_t \in \mathcal{Y}$ is defined as the tuple $(o_t \otimes o_t, \psi_{t+1} \otimes o_t)$. Let $\mathbf{\Psi} \in \mathcal{X} \otimes \mathbb{R}^N, \mathbf{\Xi} \in \mathcal{Y} \otimes \mathbb{R}^N$ and $\mathbf{H} \in \mathcal{Z} \otimes \mathbb{R}^N$ be operators that represent training data. Specifically, $\psi_s, \xi_s, h_s$ are the $s^{th}$ "columns" in $\mathbf{\Psi}$ and $\mathbf{\Xi}$ and $\mathbf{H}$ respectively. It is possible to implement S1 using a non-parametric regression method that takes the form of a linear smoother. In such case the training data for S2 regression take the form

$$\hat{\mathbb{E}}[\psi_t \mid h_t] = \sum_{s=1}^{N} \beta_{s|h_t} \psi_s$$

$$\hat{\mathbb{E}}[\xi_t \mid h_t] = \sum_{s=1}^{N} \gamma_{s|h_t} \xi_s,$$

where $\beta_s$ and $\gamma_s$ depend on $h_t$. This produces the following training operators for S2 regression:

$$\tilde{\mathbf{\Psi}} = \mathbf{\Psi}\mathbf{B}$$

$$\tilde{\mathbf{\Xi}} = \mathbf{\Xi}\mathbf{\Gamma},$$

where $\mathbf{B}_{st} = \beta_{s|h_t}$ and $\mathbf{\Gamma}_{st} = \gamma_{s|h_t}$. With this data, S2 regression uses a Gram matrix formulation to estimate the operator

$$W = \mathbf{\Xi}\mathbf{\Gamma}(\mathbf{B}^\top G_{\mathcal{X},\mathcal{X}}\mathbf{B} + \lambda I_N)^{-1}\mathbf{B}^\top \mathbf{\Psi}^* \tag{A.11}$$

Note that we can use an arbitrary method to estimate $\mathbf{B}$. Using conditional mean maps, the weight matrix $\mathbf{B}$ is computed using kernel ridge regression

$$\mathbf{B} = (G_{\mathcal{Z},\mathcal{Z}} + \lambda I_N)^{-1}G_{\mathcal{Z},\mathcal{Z}} \tag{A.12}$$

HSE-PSR learning is similar to this setting, with $\psi_t$ being a conditional expectation operator of test observations given test actions. For this reason, kernel ridge regression is replaced by application of kernel Bayes rule [6].

For each $t$, S1 regression will produce a denoised prediction $\hat{E}[\xi_t \mid h_t]$ as a linear combination of training feature maps

$$\hat{E}[\xi_t \mid h_t] = \mathbf{\Xi}\alpha_t = \sum_{s=1}^{N} \alpha_{t,s}\xi_s$$

This corresponds to the covariance operators

$$\hat{\Sigma}_{\psi_{t+1}o_t|h_t} = \sum_{s=1}^{N} \alpha_{t,s}\psi_{s+1} \otimes o_s = \mathbf{\Psi}'\text{diag}(\alpha_t)\mathbf{O}^*$$

$$\hat{\Sigma}_{o_t o_t|h_t} = \sum_{s=1}^{N} \alpha_{t,s}o_s \otimes o_s = \mathbf{O}\text{diag}(\alpha_t)\mathbf{O}^*$$

Where, $\mathbf{\Psi}'$ is the shifted future training operator satisfying $\mathbf{\Psi}'e_t = \psi_{t+1}$ Given these two covariance operators, we can use kernel Bayes rule [6] to condition on $o_t$ which gives

$$q_{t+1} = \hat{E}[\psi_{t+1} \mid h_t] = \hat{\Sigma}_{\psi_{t+1}o_t|h_t}(\hat{\Sigma}_{o_t o_t|h_t} + \lambda I)^{-1}o_t. \tag{A.13}$$

Replacing $o_t$ in (A.13) with its conditional expectation $\sum_{s=1}^{N} \alpha_s o_s$ corresponds to marginalizing over $o_t$ (i.e. prediction). A stable Gram matrix formulation for (A.13) is given by [6]

$$
\begin{aligned}
q_{t+1} \\
&= \mathbf{\Psi}' \mathrm{diag}(\alpha_t) G_{\mathcal{O},\mathcal{O}} ((\mathrm{diag}(\alpha_t) G_{\mathcal{O},\mathcal{O}})^2 + \lambda N I)^{-1} \\
&\quad .\mathrm{diag}(\alpha_t) \mathbf{O}^* o_{t+1} \\
&= \mathbf{\Psi}' \tilde{\alpha}_{t+1},
\end{aligned}
\tag{A.14}
$$

which is the state update equation in HSE-PSR. Given $\tilde{\alpha}_{t+1}$ we perform S2 regression to estimate

$$
\hat{P}_{t+1} = \hat{\mathbb{E}}[\xi_{t+1} \mid o_{1:t+1}] = \mathbf{\Xi} \alpha_{t+1} = W \mathbf{\Psi}' \tilde{\alpha}_{t+1},
$$

where $W$ is defined in (A.11).

## B Proofs

### B.1 Proof of Main Theorem

In this section we provide a proof for theorem 2. We provide finite sample analysis of the effects of S1 regression, covariance estimation and regularization. The asymptotic statement becomes a natural consequence.

We will make use of matrix Bernstein's inequality stated below:

**Lemma B.1** (Matrix Bernstein's Inequality [7]). *Let A be a random square symmetric matrix, and $r > 0$, $v > 0$ and $k > 0$ be such that, almost surely,*

$$
\mathbb{E}[A] = 0, \quad \lambda_{\max}[A] \leq r,
$$
$$
\lambda_{\max}[\mathbb{E}[A^2]] \leq v, \quad \mathrm{tr}(\mathbb{E}[A^2]) \leq k.
$$

*If $A_1, A_2, \ldots, A_N$ are independent copies of A, then for any $t > 0$,*

$$
\Pr\left[ \lambda_{\max} \left[ \frac{1}{N} \sum_{t=1}^{N} A_t \right] > \sqrt{\frac{2vt}{N}} + \frac{rt}{3N} \right]
$$
$$
\leq \frac{kt}{v}(e^t - t - 1)^{-1}.
\tag{B.1}
$$

*If $t \geq 2.6$, then $t(e^t - t - 1)^{-1} \leq e^{-t/2}$.*

Recall that, assuming $x_{test} \in \mathcal{R}(\Sigma_{\bar{x}\bar{x}})$, we have three sources of error: first, the error in S1 regression causes the input to S2 regression procedure $(\hat{x}_t, \hat{y}_t)$ to be a perturbed version of the true $(\bar{x}_t, \bar{y}_t)$; second, the covariance operators are estimated from a finite sample of size $N$; and third, there is the effect of regularization. In the proof, we characterize the effect of each source of error. To do so, we define the following intermediate quantities:

$$
W_\lambda = \Sigma_{\bar{y}\bar{x}} (\Sigma_{\bar{x}\bar{x}} + \lambda I)^{-1}
\tag{B.2}
$$
$$
\bar{W}_\lambda = \hat{\Sigma}_{\bar{y}\bar{x}} \left( \hat{\Sigma}_{\bar{x}\bar{x}} + \lambda I \right)^{-1},
\tag{B.3}
$$

where

$$
\hat{\Sigma}_{\bar{y}\bar{x}} \equiv \frac{1}{N} \sum_{t=1}^{N} \bar{y}_t \otimes \bar{x}_t
$$

and $\hat{\Sigma}_{\bar{x}\bar{x}}$ is defined similarly. Basically, $W_\lambda$ captures only the effect of regularization and $\bar{W}_\lambda$ captures in addition the effect of finite sample estimate of the covariance. $\bar{W}_\lambda$ is the result of S2 regression if $\bar{x}$ and $\bar{y}$ were perfectly recovered by S1 regression. It is important to note that $\hat{\Sigma}_{\bar{x}\bar{y}}$ and $\hat{\Sigma}_{\bar{x}\bar{x}}$ are *not* observable quantities since they depend on the true expectations $\bar{x}$ and $\bar{y}$. We will use $\lambda_{xi}$ and $\lambda_{yi}$ to denote the $i^{th}$ eigenvalue of $\Sigma_{\bar{x}\bar{x}}$ and $\Sigma_{\bar{y}\bar{y}}$ respectively in descending order and we will use $\|.\|$ to denote the operator norm.

Before we prove the main theorem, we define the quantities $\zeta_{\delta,N}^{\bar{x}\bar{x}}$ and $\zeta_{\delta,N}^{\bar{x}\bar{y}}$ which we use to bound the effect of covariance estimation from finite data, as stated in the following lemma:

**Lemma B.2** (Covariance error bound). *Let $N$ be a positive integer and $\delta \in (0, 1)$ and assume that $\|\bar{x}\|, \|\bar{y}\| < c < \infty$ almost surely. Let $\zeta_{\delta,N}^{\bar{x}\bar{y}}$ be defined as:*

$$\zeta_{\delta,N}^{\bar{x}\bar{y}} = \sqrt{\frac{2vt}{N}} + \frac{rt}{3N}, \tag{B.4}$$

*where*

$$t = \max(2.6, 2\log(4k/\delta v))$$
$$r = c^2 + \|\Sigma_{\bar{y}\bar{x}}\|$$
$$v = c^2 \max(\lambda_{y1}, \lambda_{x1}) + \|\Sigma_{\bar{x}\bar{y}}\|^2$$
$$k = c^2(\operatorname{tr}(\Sigma_{\bar{x}\bar{x}}) + \operatorname{tr}(\Sigma_{\bar{y}\bar{y}}))$$

*In addition, let $\zeta_{\delta,N}^{\bar{x}\bar{x}}$ be defined as:*

$$\zeta_{\delta,N}^{\bar{x}\bar{x}} = \sqrt{\frac{2v't'}{N}} + \frac{r't'}{3N}, \tag{B.5}$$

*where*

$$t' = \max(2.6, 2\log(4k'/\delta v'))$$
$$r' = c^2 + \lambda_{x1}$$
$$v' = c^2 \lambda_{x1} + \lambda_{x1}^2$$
$$k' = c^2\operatorname{tr}(\Sigma_{\bar{x}\bar{x}})$$

*and define $\zeta_{\delta,N}^{\bar{y}\bar{y}}$ similarly for $\Sigma_{\bar{y}\bar{y}}$.*

*It follows that, with probability at least $1 - \delta/2$,*

$$\|\hat{\Sigma}_{\bar{y}\bar{x}} - \Sigma_{\bar{y}\bar{x}}\| < \zeta_{\delta,N}^{\bar{x}\bar{y}}$$
$$\|\hat{\Sigma}_{\bar{x}\bar{x}} - \Sigma_{\bar{x}\bar{x}}\| < \zeta_{\delta,N}^{\bar{x}\bar{x}}$$
$$\|\hat{\Sigma}_{\bar{y}\bar{y}} - \Sigma_{\bar{y}\bar{y}}\| < \zeta_{\delta,N}^{\bar{y}\bar{y}}$$

*Proof.* We show that each statement holds with probability at least $1 - \delta/6$. The claim then follows directly from the union bound. We start with $\zeta_{\delta,N}^{\bar{x}\bar{x}}$. By setting $A_t = \bar{x}_t \otimes \bar{x}_t - \Sigma_{\bar{x}\bar{x}}$ then we would like to obtain a high probability bound on $\|\frac{1}{N}\sum_{t=1}^N A_t\|$. Lemma B.1 shows that, in order to satisfy the bound with probability at least $1 - \delta/6$, it suffices to set $t$ to $\max(2.6, 2k\log(6/\delta v))$. So, it remains to find suitable values for $r, v$ and $k$:

$$\lambda_{\max}[A] \leq \|\bar{x}\|^2 + \|\Sigma_{\bar{x}\bar{x}}\| \leq c^2 + \lambda_{x1} = r'$$
$$\lambda_{\max}[\mathbb{E}[A^2]] = \lambda_{\max}[\mathbb{E}[\|\bar{x}\|^2(\bar{x}\otimes\bar{x}) - (\bar{x}\otimes\bar{x})\Sigma_{\bar{x}\bar{x}} - \Sigma_{\bar{x}\bar{x}}(\bar{x}\otimes\bar{x}) + \Sigma_{\bar{x}\bar{x}}{}^2]]$$
$$= \lambda_{\max}[\mathbb{E}[\|\bar{x}\|^2(\bar{x}\otimes\bar{x}) - \Sigma_{\bar{x}\bar{x}}{}^2]] \leq c^2\lambda_{x1} + \lambda_{x1}^2 = v'$$
$$\operatorname{tr}[\mathbb{E}[A^2]] = \operatorname{tr}[\mathbb{E}[\|\bar{x}\|^2(\bar{x}\otimes\bar{x}) - \Sigma_{\bar{x}\bar{x}}{}^2]] \leq \operatorname{tr}[\mathbb{E}[\|\bar{x}\|^2(\bar{x}\otimes\bar{x})]] \leq c^2\operatorname{tr}(\Sigma_{\bar{x}\bar{x}}) = k'$$

The case of $\zeta_{\delta,N}^{\bar{y}\bar{y}}$ can be proven similarly. Now moving to $\zeta_{\delta,N}^{\bar{x}\bar{y}}$, we have $B_t = \bar{y}_t \otimes \bar{x}_t - \Sigma_{\bar{y}\bar{x}}$. Since $B_t$ is not square, we use the Hermitian dilation $\mathscr{H}(B)$ defined as follows[8]:

$$A = \mathscr{H}(B) = \begin{bmatrix} 0 & B \\ B^* & 0 \end{bmatrix}$$

Note that

$$\lambda_{\max}[A] = \|B\|, \quad A^2 = \begin{bmatrix} BB^* & 0 \\ 0 & B^*B \end{bmatrix}$$

therefore suffices to bound $\|\frac{1}{N}\sum_{t=1}^N A_t\|$ using an argument similar to that used in $\zeta_{\delta,N}^{\bar{x}\bar{x}}$ case. $\square$

To prove theorem 2, we write

$$\|\hat{W}_\lambda x_{\text{test}} - W x_{\text{test}}\|_{\mathcal{Y}} \leq \|(\hat{W}_\lambda - \bar{W}_\lambda)\bar{x}_{\text{test}}\|_{\mathcal{Y}}$$
$$+ \|(\bar{W}_\lambda - W_\lambda)\bar{x}_{\text{test}}\|_{\mathcal{Y}}$$
$$+ \|(W_\lambda - W)\bar{x}_{\text{test}}\|_{\mathcal{Y}} \tag{B.6}$$

We will now present bounds on each term. We consider the case where $\bar{x}_{\text{test}} \in \mathcal{R}(\Sigma_{\bar{x}\bar{x}})$. Extension to $\overline{\mathcal{R}(\Sigma_{\bar{x}\bar{x}})}$ is a result of the assumed boundedness of $W$, which implies the boundedness of $\hat{W}_\lambda - W$.

**Lemma B.3** (Error due to S1 Regression). *Assume that $\|\bar{x}\|, \|\bar{y}\| < c < \infty$ almost surely, and let $\eta_{\delta,N}$ be as defined in Definition 1. The following holds with probability at least $1 - \delta$*

$$\|\hat{W}_\lambda - \bar{W}_\lambda\| \leq \sqrt{\lambda_{y1} + \zeta_{\delta,N}^{\bar{y}\bar{y}}} \frac{(2c\eta_{\delta,N} + \eta_{\delta,N}{}^2)}{\lambda^{\frac{3}{2}}}$$
$$+ \frac{(2c\eta_{\delta,N} + \eta_{\delta,N}{}^2)}{\lambda}$$
$$= O\left(\eta_{\delta,N}\left(\frac{1}{\lambda} + \frac{\sqrt{1 + \frac{\log(1/\delta)}{\sqrt{N}}}}{\lambda^{\frac{3}{2}}}\right)\right).$$

*The asymptotic statement assumes $\eta_{\delta,N} \to 0$ as $N \to \infty$.*

*Proof.* Write $\hat{\Sigma}_{\hat{x}\hat{x}} = \hat{\Sigma}_{\bar{x}\bar{x}} + \Delta_x$ and $\hat{\Sigma}_{\hat{y}\hat{x}} = \hat{\Sigma}_{\bar{y}\bar{y}}x + \Delta_{yx}$. We know that, with probability at least $1 - \delta/2$, the following is satisfied for all unit vectors $\phi_x \in \mathcal{X}$ and $\phi_y \in \mathcal{Y}$

$$\langle \phi_y, \Delta_{yx}\phi_x\rangle_{\mathcal{Y}} = \frac{1}{N}\sum_{t=1}^N \langle \phi_y, \hat{y}_t\rangle_{\mathcal{Y}}\langle \phi_x, \hat{x}_t\rangle_{\mathcal{X}}$$
$$- \langle \phi_y, \hat{y}_t\rangle_{\mathcal{Y}}\langle \phi_x, \bar{x}_t\rangle_{\mathcal{X}}$$
$$+ \langle \phi_y, \hat{y}_t\rangle_{\mathcal{Y}}\langle \phi_x, \bar{x}_t\rangle_{\mathcal{X}} - \langle \phi_y, \bar{y}_t\rangle_{\mathcal{Y}}\langle \phi_x, \bar{x}_t\rangle_{\mathcal{X}}$$
$$= \frac{1}{N}\sum_t \langle \phi_y, \bar{y}_t + (\hat{y}_t - \bar{y}_t)\rangle_{\mathcal{Y}}\langle \phi_x, \hat{x}_t - \bar{x}_t\rangle_{\mathcal{X}}$$
$$+ \langle \phi_y, \hat{y}_t - \bar{y}_t\rangle_{\mathcal{Y}}\langle \phi_x, \bar{x}_t\rangle_{\mathcal{X}}$$
$$\leq 2c\eta_{\delta,N} + \eta_{\delta,N}^2$$

Therefore,

$$\|\Delta_{yx}\| = \sup_{\|\phi_x\|_{\mathcal{X}}\leq 1, \|\phi_y\|_{\mathcal{Y}}\leq 1} \langle \phi_y, \Delta_{yx}\phi_x\rangle_{\mathcal{Y}} \leq 2c\eta_{\delta,N} + \eta_{\delta,N}^2,$$

and similarly

$$\|\Delta_x\| \leq 2c\eta_{\delta,N} + \eta_{\delta,N}{}^2,$$

with probability $1 - \delta/2$. We can write

$$\hat{W}_\lambda - \bar{W}_\lambda = \hat{\Sigma}_{\bar{y}\bar{x}}\left((\hat{\Sigma}_{\bar{x}\bar{x}} + \Delta_x + \lambda I)^{-1} - (\hat{\Sigma}_{\bar{x}\bar{x}} + \lambda I)^{-1}\right)$$
$$+ \Delta_{yx}(\hat{\Sigma}_{\bar{x}\bar{x}} + \Delta_x + \lambda I)^{-1}$$

Using the fact that $B^{-1} - A^{-1} = B^{-1}(A - B)A^{-1}$ for invertible operators $A$ and $B$ we get

$$\hat{W}_\lambda - \bar{W}_\lambda = -\hat{\Sigma}_{\bar{y}\bar{x}}(\hat{\Sigma}_{\bar{x}\bar{x}} + \lambda I)^{-1}\Delta_x(\hat{\Sigma}_{\bar{x}\bar{x}} + \Delta_x + \lambda I)^{-1}$$
$$+ \Delta_{yx}(\hat{\Sigma}_{\bar{x}\bar{x}} + \Delta_x + \lambda I)^{-1}$$

we then use the decomposition $\hat{\Sigma}_{\bar{y}\bar{x}} = \hat{\Sigma}_{\bar{y}\bar{y}}^{\frac{1}{2}}V\hat{\Sigma}_{\bar{x}\bar{x}}^{\frac{1}{2}}$, where $V$ is a correlation operator satisfying $\|V\| \leq 1$. This gives

$$\hat{W}_\lambda - \bar{W}_\lambda =$$
$$- \hat{\Sigma}_{\bar{y}\bar{y}}^{\frac{1}{2}}V\hat{\Sigma}_{\bar{x}\bar{x}}^{\frac{1}{2}}(\hat{\Sigma}_{\bar{x}\bar{x}} + \lambda I)^{-\frac{1}{2}}(\hat{\Sigma}_{\bar{x}\bar{x}} + \lambda I)^{-\frac{1}{2}}\Delta_x(\hat{\Sigma}_{\bar{x}\bar{x}} + \Delta_x + \lambda I)^{-1}$$
$$+ \Delta_{yx}(\hat{\Sigma}_{\bar{x}\bar{x}} + \Delta_x + \lambda I)^{-1}$$

Noting that $\|\hat{\Sigma}_{\bar{x}\bar{x}}^{\frac{1}{2}}(\hat{\Sigma}_{\bar{x}\bar{x}} + \lambda I)^{-\frac{1}{2}}\| \le 1$, the rest of the proof follows from triangular inequality and the fact that $\|AB\| \le \|A\|\|B\|$ $\qquad\square$

**Lemma B.4** (Error due to Covariance). *Assuming that $\|\bar{x}\|_{\mathcal{X}}, \|\bar{y}\|_{\mathcal{Y}} < c < \infty$ almost surely, the following holds with probability at least $1 - \frac{\delta}{2}$*

$$\|\bar{W}_\lambda - W_\lambda\| \le \sqrt{\lambda_{y1}}\zeta_{\delta,N}^{\bar{x}\bar{x}}\lambda^{-\frac{3}{2}} + \frac{\zeta_{\delta,N}^{\bar{x}\bar{y}}}{\lambda}$$

*, where $\zeta_{\delta,N}^{\bar{x}\bar{x}}$ and $\zeta_{\delta,N}^{\bar{x}\bar{y}}$ are as defined in Lemma B.2.*

*Proof.* Write $\hat{\Sigma}_{\bar{x}\bar{x}} = \Sigma_{\bar{x}\bar{x}} + \Delta_x$ and $\hat{\Sigma}_{\bar{y}\bar{x}} = \Sigma_{\bar{y}\bar{x}} + \Delta_{yx}$. Then we get

$$\bar{W}_\lambda - W_\lambda = \Sigma_{\bar{y}\bar{x}}\left((\Sigma_{\bar{x}\bar{x}} + \Delta_x + \lambda I)^{-1} - (\Sigma_{\bar{x}\bar{x}} + \lambda I)^{-1}\right) + \Delta_{yx}(\Sigma_{\bar{x}\bar{x}} + \Delta_x + \lambda I)^{-1}$$

Using the fact that $B^{-1} - A^{-1} = B^{-1}(A - B)A^{-1}$ for invertible operators $A$ and $B$ we get

$$\bar{W}_\lambda - W_\lambda = -\Sigma_{\bar{y}\bar{x}}(\Sigma_{\bar{x}\bar{x}} + \lambda I)^{-1}\Delta_x(\Sigma_{\bar{x}\bar{x}} + \Delta_x + \lambda I)^{-1} + \Delta_{yx}(\Sigma_{\bar{x}\bar{x}} + \Delta_x + \lambda I)^{-1}$$

we then use the decomposition $\Sigma_{\bar{y}\bar{x}} = \Sigma_{\bar{y}\bar{y}}^{\frac{1}{2}}V\Sigma_{\bar{x}\bar{x}}^{\frac{1}{2}}$, where $V$ is a correlation operator satisfying $\|V\| \le 1$. This gives

$$\bar{W}_\lambda - W_\lambda =$$
$$-\Sigma_{\bar{y}\bar{y}}^{\frac{1}{2}}V\Sigma_{\bar{x}\bar{x}}^{\frac{1}{2}}(\Sigma_{\bar{x}\bar{x}} + \lambda I)^{-\frac{1}{2}}(\Sigma_{\bar{x}\bar{x}} + \lambda I)^{-\frac{1}{2}}$$
$$\cdot\Delta_x(\Sigma_{\bar{x}\bar{x}} + \Delta_x + \lambda I)^{-1}$$
$$+ \Delta_{yx}(\Sigma_{\bar{x}\bar{x}} + \Delta_x + \lambda I)^{-1}$$

Noting that $\|\Sigma_{\bar{x}\bar{x}}^{\frac{1}{2}}(\Sigma_{\bar{x}\bar{x}} + \lambda I)^{-\frac{1}{2}}\| \le 1$, the rest of the proof follows from triangular inequality and the fact that $\|AB\| \le \|A\|\|B\|$ $\qquad\square$

**Lemma B.5** (Error due to Regularization on inputs within $\mathcal{R}(\Sigma_{\bar{x}\bar{x}})$). *For any $x \in \mathcal{R}(\Sigma_{\bar{x}\bar{x}})$ s.t. $\|x\|_{\mathcal{X}} \le 1$ and $\|\Sigma_{\bar{x}\bar{x}}^{-\frac{1}{2}}x\|_{\mathcal{X}} \le C$. The following holds*

$$\|(W_\lambda - W)x\|_{\mathcal{Y}} \le \frac{1}{2}\sqrt{\lambda}\|W\|_{HS}C$$

*Proof.* Since $x \in \mathcal{R}(\Sigma_{\bar{x}\bar{x}}) \subseteq \mathcal{R}(\Sigma_{\bar{x}\bar{x}}^{\frac{1}{2}})$, we can write $x = \Sigma_{\bar{x}\bar{x}}^{\frac{1}{2}}v$ for some $v \in \mathcal{X}$ s.t. $\|v\|_{\mathcal{X}} \le C$. Then

$$(W_\lambda - W)x = \Sigma_{\bar{y}\bar{x}}((\Sigma_{\bar{x}\bar{x}} + \lambda I)^{-1} - \Sigma_{\bar{x}\bar{x}}^{-1})\Sigma_{\bar{x}\bar{x}}^{\frac{1}{2}}v$$

Let $D = \Sigma_{\bar{y}\bar{x}}((\Sigma_{\bar{x}\bar{x}} + \lambda I)^{-1} - \Sigma_{\bar{x}\bar{x}}^{-1})\Sigma_{\bar{x}\bar{x}}^{\frac{1}{2}}$. We will bound the Hilbert-Schmidt norm of $D$. Let $\psi_{xi} \in \mathcal{X}$, $\psi_{yi} \in \mathcal{Y}$ denote the eigenvector corresponding to $\lambda_{xi}$ and $\lambda_{yi}$ respectively. Define $s_{ij} = |\langle\psi_{yj}, \Sigma_{\bar{x}\bar{y}}\psi_{xi}\rangle_{\mathcal{Y}}|$. Then we have

$$|\langle\psi_{yj}, D\psi_{xi}\rangle_{\mathcal{Y}}| = \left|\langle\psi_{yj}, \Sigma_{\bar{y}\bar{x}}\frac{\lambda}{(\lambda_{xi} + \lambda)\sqrt{\lambda_{xi}}}\psi_{xi}\rangle_{\mathcal{Y}}\right|$$
$$= \frac{\lambda s_{ij}}{(\lambda_{xi} + \lambda)\sqrt{\lambda_{xi}}} = \frac{s_{ij}}{\sqrt{\lambda_{xi}}}\frac{1}{\frac{1}{\lambda/\lambda_{xi}} + 1}$$
$$\le \frac{s_{ij}}{\sqrt{\lambda_{xi}}}\cdot\frac{1}{2}\sqrt{\frac{\lambda}{\lambda_{xi}}} = \frac{1}{2}\sqrt{\lambda}\frac{s_{ij}}{\lambda_{xi}}$$
$$= \frac{1}{2}\sqrt{\lambda}|\langle\psi_{yj}, W\psi_{xi}\rangle_{\mathcal{Y}}|,$$

where the inequality follows from the arithmetic-geometric-harmonic mean inequality. This gives the following bound

$$\|D\|_{HS}^2 = \sum_{i,j} \langle \psi_{yj}, D\psi_{xi}\rangle_{\mathcal{Y}}^2 \le \frac{1}{2}\sqrt{\lambda}\|W\|_{HS}^2$$

and hence

$$\|(W_\lambda - W)x\|_{\mathcal{Y}} \le \|D\|\|v\|_{\mathcal{X}} \le \|D\|_{HS}\|v\|_{\mathcal{X}}$$
$$\le \frac{1}{2}\sqrt{\lambda}\|W\|_{HS}C$$

$\square$

Note that the additional assumption that $\|\Sigma_{\bar{x}\bar{x}}^{-\frac{1}{2}}x\|_{\mathcal{X}} \le C$ is not required to obtain an asymptotic $O(\sqrt{\lambda})$ rate for a given $x$. This assumption, however, allows us to uniformly bound the constant. Theorem 2 is simply the result of plugging the bounds in Lemmata B.3, B.4, and B.5 into (B.6) and using the union bound.

### B.2  Proof of Lemma 3

for $t = 1$: Let $\mathcal{I}$ be an index set over training instances such that

$$\hat{q}_1^{\text{test}} = \frac{1}{|\mathcal{I}|}\sum_{i \in \mathcal{I}} \hat{q}_i$$

Then

$$\|\hat{q}_1^{\text{test}} - \tilde{q}_1^{\text{test}}\|_{\mathcal{X}} = \frac{1}{|\mathcal{I}|}\sum_{i \in \mathcal{I}} \|\hat{q}_i - \tilde{q}_i\|_{\mathcal{X}} \le \frac{1}{|\mathcal{I}|}\sum_{i \in \mathcal{I}} \|\hat{q}_i - q_i\|_{\mathcal{X}} \le \eta_{\delta,N}$$

for $t > 1$: Let $A$ denote a projection operator on $\mathcal{R}^{\perp}(\Sigma_{\bar{y}\bar{y}})$

$$\|\hat{q}_{t+1}^{\text{test}} - \tilde{q}_{t+1}^{\text{test}}\|_{\mathcal{X}} \le L\|\hat{p}_t^{\text{test}} - \tilde{p}_t^{\text{test}}\|_{\mathcal{Y}} \le L\|A\hat{W}_\lambda \hat{q}_t^{\text{test}}\|_{\mathcal{Y}}$$
$$\le L\left\|\frac{1}{N}\left(\sum_{i=1}^N A\hat{p}_i \otimes \hat{q}_i\right)\left(\frac{1}{N}\sum_{i=1}^N \hat{q}_i \otimes \hat{q}_i + \lambda I\right)^{-1}\right\|\|\hat{q}_t^{\text{test}}\|_{\mathcal{X}}$$
$$\le L\left\|\frac{1}{N}\sum_{i=1}^N A\hat{p}_i \otimes A\hat{p}_i\right\|^{\frac{1}{2}}\frac{1}{\sqrt{\lambda}}\|\hat{q}_t^{\text{test}}\|_{\mathcal{X}} \le L\frac{\eta_{\delta,N}}{\sqrt{\lambda}}\|\hat{q}_t^{\text{test}}\|_{\mathcal{X}},$$

where the second to last inequality follows from the decomposition similar to $\Sigma_{YX} = \Sigma_Y^{\frac{1}{2}}V\Sigma_X^{\frac{1}{2}}$, and the last inequality follows from the fact that $\|A\hat{p}_i\|_{\mathcal{Y}} \le \|\hat{p}_i - \bar{p}_i\|_{\mathcal{Y}}$. $\square$

## C  Examples of S1 Regression Bounds

The following propositions provide concrete examples of S1 regression bounds $\eta_{\delta,N}$ for practical regression models.

**Proposition C.1.** *Assume* $\mathcal{X} \equiv \mathbb{R}^{d_x}, \mathbb{R}^{d_y}, \mathbb{R}^{d_z}$ *for some* $d_x, d_y, d_z < \infty$ *and that* $\bar{x}$ *and* $\bar{y}$ *are linear vector functions of* $z$ *where the parameters are estimated using ordinary least squares. Assume that* $\|\bar{x}\|_{\mathcal{X}}, \|\bar{y}\|_{\mathcal{Y}} < c < \infty$ *almost surely. Let* $\eta_{\delta,N}$ *be as defined in Definition 1. Then*

$$\eta_{\delta,N} = O\left(\sqrt{\frac{d_z}{N}}\log((d_x + d_y)/\delta)\right)$$

*Proof.* (sketch) This is based on results that bound parameter estimation error in linear regression with univariate response (e.g. [9]). Note that if $\bar{x}_{ti} = U_i^\top z_t$ for some $U_i \in \mathcal{Z}$, then a bound on the error norm $\|\hat{U}_i - U_i\|$ implies a uniform bound of the same rate on $\hat{x}_i - \bar{x}$. The probability of exceeding the bound is scaled by $1/(d_x + d_y)$ to correct for multiple regressions. □

Variants of Proposition C.1 can also be developed using bounds on non-linear regression models (e.g., generalized linear models).

The next proposition addresses a scenario where $\mathcal{X}$ and $\mathcal{Y}$ are infinite dimensional.

**Proposition C.2.** *Assume that $x$ and $y$ are kernel evaluation functionals, $\bar{x}$ and $\bar{y}$ are linear vector functions of $z$ where the linear operator is estimated using conditional mean embedding [4] with regularization parameter $\lambda_0 > 0$ and that $\|\bar{x}\|_\mathcal{X}, \|\bar{y}\|_\mathcal{Y} < c < \infty$ almost surely. Let $\eta_{\delta,N}$ be as defined in Definition 1. It follows that*

$$\eta_{\delta,N} = O\left(\sqrt{\lambda_0} + \sqrt{\frac{\log(N/\delta)}{\lambda_0 N}}\right)$$

*Proof.* (sketch) This bound is based on [4], which gives a bound on the error in estimating the conditional mean embedding. The error probability is adjusted by $\delta/4N$ to accommodate the requirement that the bound holds for all training data. □

## Footnotes

[1] Following the notation used in [1], $u_x^\top \hat{P}_{2:3,1} \equiv \hat{P}_{3,x,1}$