[Reviews · NeurIPS 2015]

Submitted by Assigned_Reviewer_1

A nice advantage of predictive representations of stochastic processes is that they can be expressed in terms of families of linear operators --- the "observable operators" of Jaeger (oddly, not cited in this paper; also, see Upper, and the appendix to Shalizi and Crutchfield). This paper proposes (following some earlier work) to exploit this fact, by using the instrumental variables technique from econometrics to simplify the estimation of such models.

Doing so results in an estimation procedure very similar to that of Langford et al. from 2009 (reference [16] in the paper), but with some advantages in terms of avoiding iterative re-estimation. However, there seems to be an important issue which isn't (that I saw) addressed here.

The instrumental variable needs to be correlated with the input variable to the regression, but independent of the noise in the regression.

The instruments used here are features of the past observations.

The noise in future observations, and even future predictive states, has however a distribution which in general changes with those past observations.

It is thus not at all clear to me that the IV technique really is valid here, or rather under what conditions it is valid. I must, however, admit that I have not had the time to carefully trace through the proofs, especially in the supplementary material, so it may well be that this concern is addressed, or that there is a way to see that it's no cause for concern.

@article{Jaeger-operator-models,

author = "Herbert Jaeger",

title = "Observable Operator Models for Discrete Stochastic Time Series",

url = "http://minds.jacobs-university.de/sites/default/files/uploads/papers/oom_neco00.pdf",

journal = "Neural Computation",

volume = 12,

pages = "1371--1398",

year = 2000,

doi = "10.1162/089976600300015411"}

@article{Shalizi-Crutchfield-cmech,

author = "Cosma Rohilla Shalizi and James P. Crutchfield",

title = "Computational Mechanics: Pattern and Prediction, Structure and Simplicity",

journal = "Journal of Statistical Physics",

volume = 104,

year = 2001,

pages = "817--879",

URL = "http://arxiv.org/abs/cond-mat/9907176"}

@PhdThesis{Upper-thesis,

author = "Daniel R. Upper",

title = "Theory and Algorithms for Hidden {Markov} Models and Generalized Hidden {Markov} Models",

year = 1997,

URL = "http://csc.ucdavis.edu/~cmg/compmech/pubs/TAHMMGHMM.htm",

school = "University of California, Berkeley"}
Summary: This paper proposes to improve estimation for predictive representations of stochastic processes by means of the instrumental variable technique.

It's an ingenious idea, but I am not quite sure of its validity.

Submitted by Assigned_Reviewer_2

Summary -------

The authors present a new algorithm for learning dynamical systems, which generalizes some existing spectral learning algorithms. The approach is based on keeping track of the latent state via a observable representation, which can be updated using a linear mapping for the model classes consider in the manuscript (eg HMMs, linear dynamical systems). To arrive at an unbiased estimate of this mapping from data the authors apply instrumental variable regression.

Comments --------

The article contains interesting contributions to the literature on learning dynamical systems. Basic versions of the idea of formulating dynamical system learning as supervised learning problems have be, but the paper gives useful generalizations of the approach and proves error bounds. The main drawbacks are the relative simple numerical experiments and that the manuscript feels a little hastily written. The latter might have contributed to me understanding the approach less well than I would like to.

- A main advantage of the proposed approach, that the authors point out, is that the formulation of the learning procedure as regression problems allows to tap into the regression literature for improving and generalizing the algorithm. However, I don't understand why this should mostly apply to the stage 1 regression. E.g. the experiment presented in 5.2: I'm guessing in this case the $W$ is closely related to the transition matrix $T$. Wouldn't it make sense to try to learn a structured $W$ that somehow captures the block-structure of the prior, instead of using a L1 regression in the S1 pre-processing step. Also, in a non-linearity dynamical system (say with linear Gaussian observations), would we want to use a non-linear regression for S2? What am I missing here? This should be clarified in the manuscript.

- Sec 5.1: This a very simple model to learn. It would be much more convincing if the authors also presented a slightly more high-dimensional case. Also, how does naive EM-HMM learning perform here; this comparison would be quite instructive.

- Sec 5.1: The authors should explicitly spell out why their algorithm gives better performance in this case, and why model 3 performs better than model 2.

- Sec 5.1: Why is logistic regression a sensible choice here? This is an important demonstration for the method, so the manuscript would benefit from a more detailed explanation/justification. If the authors need more space, I suggestion possible deleting figure 1.

- The authors should cite some of the traditional system identification literature, where instrumental variables have been used for a long time, eg [Ljung. System identification, 1999] and references therein and explain where the similarities and differences lie.

- Is the regression stage S1B really necessary? If so, why?

- minor points: -- l25: missing "of"

-- l79: missing ")" -- l94: It would be useful if you could briefly remind the reader of the definition of k-observability and how this affects the choice of future feature $\psi_t$ -- l99: maybe replace "is determined" by "can be computed from" -- l121: Should it be conditioned on $o_{t-k:t+k-1}$ -- l190: typo, should prob be $P(o_{t:t+k-1}\vert o_{1:t-1})$ -- table one, top left box: there's some confusion with the indices here. For a stationary HMM, $P(o_2,o_1)$ and

$P(o_t,o_t)$ are the same, but a consistent notation should be used.
Summary: The article contains interesting contributions to the literature on learning dynamical systems. Basic versions of the idea of formulating dynamical system learning as supervised learning problems have be, but the paper gives useful generalizations of the approach and proves error bounds. The main drawbacks are the relative simple numerical experiments and that the manuscript feels a little hastily written. The latter might have contributed to me understanding the approach less well than I would like to.

Submitted by Assigned_Reviewer_3

The authors propose a generalization of recent spectral algorithms for learning dynamical system models including linear dynamical systems, hidden Markov models, and predictive state representations. The authors also provide a theoretical analysis of their approach. One of the interesting consequences of this formulation is that other forms of regression such as ridge regression (and possibly l_1 regression) can be substituted into the learning algorithm, resulting in more efficient learning algorithms. Theoretically, the paper seems very strong, however the experimental results seem weak, or at least need to be explained much better. In section 5.2.1, what does it mean to evaluate the root mean squared error of each split? The mean squared error of what?? Parameter estimates? Filtering? Predicting? I also had a lot of difficulty interpreting Figure 5.

Some additional minor comments:

Start of quotations is backward throughout. Sentence punctuation should be inside quotations.

Abstract: "...we can learn a better state representation.." -> better than what?

Intro: Citations for EM, sampling, and spectral algorithms are necessary. Which papers are the authors referring to?

I disagree that "spectral algorithms" lack an explanation based on higher level concepts than linear algebra. Many papers in the last several years have provided detailed overarching theory based on method of moments. See papers by Anandkumar, Hsu, Kakade, Liang, etc.

Section 2: Close parentheses are missing in third paragraph.

Section 2: Need to define "spectral algorithms" more precisely. A number of spectral algorithms, that is, algorithms that rely only on a singular value decomposition, can be used to recover the parameters of latent variable models and not just observable representations. There are many algorithms for learning the parameters of linear dynamical systems, as well as discrete-valued latent variable models. See recent papers by Anandkumar et al. 2012 and Shaban et al. 2015 for examples of the latter algorithms.

Section 2: The name "predictive belief" seems strange. Early in Section 2 the authors define belief as a probability distribution or density on state. This makes sense. Later in Section 2 the authors define predictive belief as an expectation of future features. This is not the standard definition of belief and is confusing.

Section 2: The sentence "So we first use regression models to estimate..." this sentence does not make sense.

Section 4: To minimize confusion, the definitions of \hat x etc in the third paragraph should be moved to somewhere before Equation 3.
Summary: The paper introduces a formal two-stage regression framework for learning a variety of dynamical system models that should be of interest to anyone who works on problems related to system identification.

Some aspects of the paper need work, including the experimental results.

Submitted by Assigned_Reviewer_4

The authors present a framework which generalizes "spectral learning" in dynamical systems. A clear exposition of a subset of recent work in the area is provided and an insightful relationship to instrumental variable regression is outlined. The outlined framework essentially generalizes by allowing for arbitrary regressions to be performed in order to "de-noise" features of history, future, and extended future. Theoretical results providing guarantees similar to previous works, but taking into account error from the "denoising" are presented, and some experiments on small (synthetic/toy) datasets are provided.

Overall, the paper was well-structured, motivated, and the theoretical analysis appears sound. It makes many ideas implicit in previous works explicit, which is a useful contribution. My main concern, however, is w.r.t. the empirical analysis. In particular, why was such a small system used for the prediction task, i.e only 2-dimensional? Moreover, its not clear why filtering and prediction is useful for this task, since this is an example where we clearly want access to the hidden state. The sparsity example is similarly underwhelming since it relies on a hand-crafted simulation and there is no real example of where/when it would be useful. Overall, this would be a strong paper if it convincingly demonstrated empirical evidence that it is useful for a compelling application. As it stands, I believe there are benefits over the previous approaches, but it is not clear if these benefits are significant for interesting problems. An additional experiment on a compelling, high-dimensional problem would be very beneficial.

Minor comments:

Since the work is presented as a generalizing framework, it needs to engage more with existing literature beyond Boots and Anandkumar. For example, the connection to weighted automata (http://dl.acm.org/citation.cfm?id=1553379) and more PSR literature (e.g., work prior to Boots et al). Balle et al 2014 (http://jmlr.org/proceedings/papers/v32/balle14.pdf) provides a recent survey and discusses a general convex-optimization based framework.

Why not also have an EM baseline? Balle et al, 2014 show that EM often strongly outperforms spectral methods.

Use `` (two single backquotes) not " (a single double quote) to get quotes to work correctly in LateX.
Summary: Well-written, clear, and insightful, this paper provides a novel and potentially useful generalization of spectral learning for dynamical systems with theoretical analysis. However, the empirical results on small/synthetic systems are not particularly compelling.

Submitted by Assigned_Reviewer_5

The paper reduces the problem of learning a dynamic system to two-stage regression, whose one special case is spectral method. In applications to HMM and Kalman filter, the proposed method can use more features and nonlinear or regularized regression and thus outperforms original spectral method.

The connections to and generalization over spectral method is interesting. But it is not very clear how to map a spectral method to the notations used in this paper.

By allowing using more features and more complicated regression model definitely provide possibilities to beat spectral method. But these might lead to more computation and thus make this method not that attractive (spectral method is attractive because it is faster than EM and can be used as initialization of EM). Comparison to traditional method such as EM algorithm is missing, so it is hard to justify the real advantages of the proposed method in practice.

Several parts of this paper are not easy to understand. The reason might be the authors tries to build a connection between their method and spectral method. But there is no one-to-one notation mapping can help readers to understand the connection.

Summary: The paper reduces the problem of learning a dynamic system to two-stage regression, whose one special case is spectral method. In applications to HMM and Kalman filter, the proposed method can use more features and nonlinear or regularized regression and thus outperforms original spectral method. The connections to and generalization over spectral method is interesting. However, it is not clear what is the advantages of this method comparing to EM based ones. Several parts of this paper are hard to understand well.

Author Feedback
Author rebuttal: We thank the reviewers for their helpful comments. We will polish the manuscript based on the reviews and include suggested references.
We would like to point out that the purpose of the first experiment is to demonstrate that in a small data setting (where spectral methods are known to have problems) and on a real-world dataset, better performance can be obtained through changing S1 step. Based on reviews, we experimented with EM and we found that two stage approach is on par with EM (slightly worse on 60% of trials) but is much faster. We are happy to include these details in the final paper.
Also, to our knowledge, this framework provides new theoretical results for models that have proven to perform well in high dimensional settings (e.g. uncontrolled HSE-PSR). Below we address other specific points:

= R1:
Sparse S2 regression (or other regularization)?
The proposed framework admits non-linear S1 regression as well as flexible regularization for S2 (we analyze the ridge regression case as it naturally applies to general RKHS but we believe the analysis can be extended to other regularization schemes).

Non-linear S2?
The linear S2 assumption is required for obtaining an unbiased estimate of W (eqn 2). This facilitates consistent learning without the need to restart the system. If we assume the ability to restart the system to the stationary distribution (so we do not need to integrate out history beyond past window), we can have non-linear S2 regression however we chose to relax this assumption. In this case, feature representation and S1 should apply the transformation needed to make S2 linear (e.g. HSE-PSR uses kernel transformation).

Why use Sparse S1 in 5.2?
In experiment 5.2 we focus on the effect of the choice of S1 regression on the obtained state representation (the basis for expressing expected future). Even if S2 regression would use lasso, incorporating the sparsity assumption in S1 regression is beneficial.

Is S2B necessary?
If we use OLS as S1 and S2 regression procedure, S1B becomes redundant. With non-linear S1, our experiments (not shown in the manuscript) have shown that S1B is indeed beneficial (intuitively it makes S2 regression learn from denoised outputs).

Why logistic regression?
We chose logistic regression since it is suitable for probabilistic output and it can use less parameters than its linear counterpart, which demonstrates how non-linearity can result in more statistically efficient models.

Predictive Belief?
We define future features to be sufficient statistics of the distribution of future observations. Thus, the expectation of these features is a probability distribution and hence a belief. Since it is a belief over observables we call it "predictive". We will make this point more clear in the final paper.

= R2:
We are sorry for confusion. By RMSE we mean that of the prediction, which is a common metric for this kind of data (see Falakmasir et al 2013).
Figure 5.2 represents the basis vectors chosen to represent predictive belief. It is desirable to have vectors whose support matches independent sub-
systems to uncover the underlying structure.

We acknowledge the existence of other classes of spectral algorithms outside the scope of this work in footnote 2. We will make this point more emphasized in the final manuscript.

= R3:
We will add discussion on the work by Balle et al. However, we would like to point out that their optimization algorithm relies on computing Hankel matrices, which can be thought of as linear S1 regression. It allows for specifying a regularizer (e.g. trace norm) when obtaining model operators. This capability is also supported by our framework (though we focused on ridge regularization in our analysis).

= R4:
We would argue that solving convex optimization problems can still be orders of magnitude faster than EM (see for example Balle et. al. suggested by R3. This was also observed when we experimented with EM as mentioned above). Also, note that having complex modeling assumptions (e.g. sparsitity) can make convex-optimization a subroutine in EM.

We are sorry for the notation mismatch but we would like to point out that there is a body of literature on spectral methods and observable operators that uses different notations. For this provided the mapping between the proposed framework and existing spectral methods on a case-by-case with a summary provided in table1 and more details in the supplementary material due to space constraints.  

= R5:
How IV applies?
We assume the noise is white (independent at each time step). With that assumption, one can think of a graphical model at time t where observed future features is a common child of future noise, past features and past noise without other dependencies. The v-structure (past features -> future observations <- future noise) implies that past observations are independent of future noise. We will add more clarification in the final paper.